# Increased RBP4 and Asprosin Are Novel Contributors in Inflammation Process of Periodontitis in Obese Rats

**DOI:** 10.3390/ijms242316739

**Published:** 2023-11-25

**Authors:** Yuwei Zhang, Yifei Zhang, Yutian Tan, Xiao Luo, Ru Jia

**Affiliations:** 1Key Laboratory of Shaanxi Province for Craniofacial Precision Medicine Research, College of Stomatology, Xi’an Jiaotong University, Xi’an 710004, China; dentistzyw0505@163.com (Y.Z.); zyf1175944673@foxmail.com (Y.Z.); 2Clinical Research Center of Shaanxi Province for Dental and Maxillofacial Diseases, Xi’an 710004, China; 3Department of Prosthodontics, College of Stomatology, Xi’an Jiaotong University, Xi’an 710004, China; 4Department of Physiology and Pathophysiology, School of Basic Medical Sciences, Xi’an Jiaotong University Health Science Center, Xi’an 710061, China; tanyutian@stu.xjtu.edu.cn; 5Department of Digital Oral Implantology and Prosthodontics, College of Stomatology, Xi’an Jiaotong University, Xi’an 710004, China

**Keywords:** obesity, periodontitis, asprosin, RBP4, adipokines

## Abstract

There is a significant comorbidity between obesity and periodontitis, while adipokines are pivotal in the immunoinflammatory process, which may play a role in this special relationship. We aimed to assess the effect of adipokines as mediators in the progression of periodontitis in obese Sprague Dawley rats. Rats were divided into four groups: normal body weight with and without periodontitis and obesity with and without periodontitis. Experimental obesity and periodontitis were induced by a high-fat diet or ligaturing, and the effect was measured using metabolic and micro-computed tomography analysis and histological staining. Compared with the other three groups, the group of periodontitis with obesity (OP) had the heaviest alveolar bone absorption, the largest increase in osteoclasts, the utmost inflammatory cell infiltration and the highest expressions of pro-inflammatory cytokines and nuclear factor-kappa B ligand (RANKL); meanwhile, its expression of the osteogenesis-related gene was the lowest among the four groups. The expressions of leptin, visfatin, resistin, retinol-binding protein 4 (RBP4) and asprosin were upregulated, while adiponectin was decreased significantly in OP. The strong positive associations between the periodontal or circulating levels of RBP4 (or asprosin) and the degree of alveolar resorption in experimental periodontitis and obese rats were revealed. The upregulated expression of inflammation biomarkers, the corresponding degradation in connective tissue and the generation of osteoclasts in periodontitis were activated and exacerbated in obesity. The elevated level of RBP4/asprosin may contribute to a more severe periodontal inflammatory state in obese rats.

## 1. Introduction

Periodontitis is a highly prevalent public health issue, and is defined by pathological loss of the dental supporting tissues during the inflammatory process. The progression of periodontitis is not only affected by periodontal local factors, but also regulated by many systemic factors, such as diabetes, smoking and so on. In recent years, many clinical studies have shown that metabolic syndrome can also affect the progression of periodontitis [1,2]. Obesity, as one of the most common symptoms of metabolic syndrome, is characterized by the excessive accumulation of adipose tissue with a state of low-grade systematic inflammation [3,4]. Studies have demonstrated the established negative relationships between obesity and bone metabolism as well as the association between obesity and periodontitis [5]. Nevertheless, the mechanism underlying obesity-exacerbated periodontium degradation is still equivocal; the linkage that associates the pathological process of obesity and periodontitis is still to be found.

Meanwhile, adipose tissue is nowadays regarded as an energy storage organ as well as a crucial endocrine organ, which produces and secretes various metabolically active molecules, including lipids, peptides, and inflammatory-related cytokines. Among the cytokines secreted by adipocytes, adipokines are expressly involved in various physiological and pathological processes, such as regulating insulin sensitivity and energy expenditure while spontaneously participating in the immunoinflammatory response and healing process [6]. Research has revealed that during the process of periodontitis, the elevated levels of several pro-inflammatory cytokines and the corresponding reduced levels after periodontal treatment have been detected both in circulating and periodontally active sites [7,8]. For the reasons above, adipokines are speculated to be potentially key factors involved in the association between obesity and periodontitis. Among these, retinol-binding protein 4 (RBP4) and asprosin are two novel adipokines with ill-defined functions in the process of obesity and periodontitis, while associations have been discovered between these two adipokines and other metabolic diseases [9] or periodontitis [10].

Thus, further assessments are demanded. Therefore, our study aimed to investigate the expression profile and possible role of RBP4, asprosin and other adipokines in experimental periodontitis and diet-induced obese rats. In addition, the potential mechanism underlying obesity-exacerbated periodontium degradation was discussed from systemic and periodontal perspectives to provide clinal pragmatic information about managing periodontal diseases in obese patients.

## 2. Results

### 2.1. Establishment of Experimental Obesity Rat Model and Related Biochemical Data

The body weight of Sprague Dawley rats (SD rats) that treated with a HFD during the 10 consecutive weeks increased dramatically and was higher than the groups treated with CHOW, with a distinguished difference (from week 4 to 12; *p* < 0.0001) at the end of week 12, reaching approximately 450 g (Figure 1A). Food intake was also recorded and we found that the incorporation of a HFD led to a decline in food consumption compared to the CHOW-fed groups during the middle–later stage of the feeding period from week 6, suggesting a compensatory reduction in the intake of fatty substances (Figure 1B). Lee’s index, as a sophisticated parameter for evaluating the degree of obese of SD rats [11], was calculated accordingly. Consistent with the results in the whole-body weight and WAT, the values of Lee’s index in the O and OP groups were significantly higher than those of the other two groups (*p* < 0.0001, Figure 1C). It is widely known and demonstrated that obesity is the accumulation of excess adiposity [12], and that brown adipose tissue (BAT), along with white adipose tissue (WAT), are the two major types of mammalian adipose tissue; hence, the weight of WAT and BAT was measured. Peripheral adipose tissue and epididymal visceral adipose tissue in the WAT are commonly recognized as the depository of excess energy in forms of triglycerides, whereas BAT is abundant in small mammals and newborns and is known for its thermogenic function in maintaining the temperature of the core body [13,14]. The quantity of weight in the representative dissection of WAT displayed a significantly higher level in groups of obesity (O) and the combination of obesity and periodontitis (OP); however, no statistical difference was discovered in BAT among the groups (Figure 1D). fasting blood glucose levels (FBGL) was measured in each group and no significant difference was found; this enabled us to exclude the cofounding factor of diabetes mellitus (5.0–9.5 mmol/L, Figure 2A). In addition, considering the level of high-density lipoprotein cholesterol (HDL-C) in plasma, only the level in OP was lower than that in the control group. Nevertheless, the concentration of triglycerides, total cholesterol and low-density lipoprotein cholesterol (LDL-C) in the plasma showed a similar trend, as the values in the groups of O and OP were dramatically higher than those in the groups of C and P (Figure 2B–D).

### 2.2. Establishment of Experimental Periodontitis Rat Model

After the 2-week ligature (Figure 3A), micro-computed tomography was applied to observe the absorption of alveolar bone tissue in rats. Sectional images of buccolingual and mesiodistal apical–coronal directions, together with the three-dimensional reconstructed micro-CT of the maxillary second molar, exhibited a distinguished difference in their quantities of maxillae resorption between any two groups. In them, the linear resorption distances from cementum–enamel junction (CEJ) to alveolar bone crest (ABC) were sorted as C, O, P and OP in an ascending order, indicating more bone loss and attachment loss in the ligated groups and the maximum bone loss in the comorbidity of obesity and periodontitis (Figure 3B,C). Consistently, Hematoxylin–eosin (H&E)-stained sections from four groups illustrated the infiltration of inflammatory cells in the periodontium (red arrowheads), the bone loss and nuance in the apical migration in the junctional epithelium of individual discrete transects. Except for the normal periodontal structure in the control group, the obesity group exhibited a slightly elevated infiltration of plasmacytes, macrophage cells and other inflammatory cells in periodontal tissue, wherein the highest infiltration levels of the aforementioned inflammatory-related cells were observed in the P and OP groups. (Figure 3D). These results confirm that the 2-week treatment of the ligature in addition to periodontal pathogens successfully induced periodontitis in rats [15,16].

### 2.3. Expression of Periodontal and Systemic Inflammatory Cytokines

Based on the aforementioned obesity and/or periodontitis rat models, we further analyzed the relative mRNA expression (regarding the control group as the baseline) of inflammatory molecules in the gingival tissue (Figure 4A). The relative expression of interleukin (IL)-1β and IL-6 presented a hierarchic ascending trend in the four groups, from the control group, obese group, and periodontitis group to the combination group, which indicated that the status of obesity or periodontitis would work as an inducement alone to trigger increased levels of the pro-inflammatory cytokines IL-1β and IL-6, while the comorbidity model would amplify such an inflammatory effect. Furthermore, the gene expression levels of tumor necrosis factor-α (TNF-α), chemokine (C-C motif) ligand 2 (CCL-2), inducible nitric oxide synthase (Nos2) and IL-17A displayed no statistical differences among the control and obese group, while the levels were elevated in the periodontitis group and combination group. IL-6, as a bio index generated transiently and promptly in response to tissue injuries and infections, contributes to host defense through stimulating the acute phase response and immune reaction [17]. And the level of IL-6 was found to increase in the periodontitis and combination groups both in gingival crevicular fluid (GCF) and plasma using enzyme-linked immunosorbent assay (ELISA) analysis, which suggests the state of inflammation, whether in periodontally active sites or as a systemic condition (Figure 4B). In the Western blot results of IL-1β in gingival tissue, as displayed in Figure 4C, the levels of it in the other three groups were significantly higher than those in the control group (Figure 4D). So far, the combined effect of obesity and periodontitis has been highlighted as the cause and magnification of inflammatory process both in the mRNA and protein levels.

### 2.4. Expression of Bone Remodeling Biomarkers in Periodontium

To observe the morphological results of the micro-remodeling of alveolar bone caused by obesity and periodontitis, tartrate-resistant acid phosphatase (TRAP) enzyme staining and related statistical analysis were performed to ascertain the presence and localization of osteoclasts. The marked TRAP-positive cells/osteoclasts in the representative images of the C, O, P and OP groups showed a gradual increase in number, and these analyses showed a significant difference between groups (*p* < 0.001). Also, the different abundance levels of osteoclasts implies the varying degrees of periodontal tissue remodeling (Figure 5A,B). Moreover, nuclear factor-kappa B ligand (RANKL), also known as TNF-sf11, is a gene expressed as a membrane-associated cytokine by osteoblasts to mediate osteoclastogenesis [18]. The maximum relative gene expression of RANKL in gingiva occurred in the OP group, followed by the P group, while no statistical difference was found between the C group and the O group (Figure 5C). Intriguingly, the opposing results showed up in the expression of alkaline phosphatase (ALP), parathyroid hormone like hormone (Pthlh), cathepsin K (Ctsk) and type I collagen (Col-1). All these genes, as well as runt-related transcription factor 2 (Runx2), are actively conducive to the process of osteogenesis or avoiding osteoclastogenesis. However, as a pivot of regulation in bone metabolism, Runx2 was expressed less in the P group, while the other three groups showed no significant difference (Figure 5D).

### 2.5. Adipokine Expression Pattern in Periodontitis and Obesity

Subsequently, we investigated the patterns of the mRNA expression of adipokine in the gingiva of the four groups and surprisingly found the upregulation of RBP4 and asprosin levels in the periodontitis group with/without the presence of obesity. Notably, the expression of RBP4 and asprosin in the OP group was significantly higher than those of the other three groups. Unlike the results in RBP4 and asprosin, obesity could independently or collectively act as an interfering factor in significantly upregulating the expression of leptin, nampt/visfatin and resistin. Considering the expression of one anti-inflammatory adipokine, adiponectin, a reverse outcome showed up, which suggests the anti-inflammatory function of adiponectin had been somehow inhibited, with a low expression in periodontitis and especially in the combination of periodontitis and obesity (Figure 6A). Western blot analysis showed that the RBP4 level was dramatically enhanced in the gingiva of periodontitis in the presence and absence of obesity. Furthermore, the levels of asprosin in the obese and/or periodontitis groups were elevated, with no statistical difference in three of the groups, not accounting for the control group. FBN1 and asprosin protein shared the same precursor of 2871 aminos acids long, encoded by FBN1 gene; further, the 140-amino-acid long C-terminal cleavage product of the precursor turned into asprosin [19]. And an increasing expression level of FBN1 was only detected in OP (Figure 6B,C). We then evaluated the circulating and GCF levels of RBP4 through ELISA analysis, and found an obesity-induced and/or periodontitis-induced elevated circulating level, with the maximum circulating RBP4 level in the OP group. When it came to the local concentration of RBP4 in GCF, the obesity and control groups did not show a statistical difference, while in the groups of P and OP it displayed a gradual improvement in concentration, with the maximum in OP. In addition, we found that the circulating level of asprosin was decreased in obese group compared with control group, which was inconsistent with the results found in several studies focusing on obese adults and mice [20,21,22]. This could be attributed to the phase of obesity. However, this reduction in the asprosin plasma level in the obesity group was offset but had a significantly increased level in the combination OP group (Figure 6D). Later, the immunohistochemistry (IHC) staining of asprosin and RBP4 were carried out to extend our observations (Figure 6E). Both the obesity and periodontitis groups had more asprosin-positive cells than the control group, with significant differences but without significant difference between these two groups, while the OP group exhibited the maximum asprosin-positive out of the other three groups statistically (Figure 6F). As for RPB4 in the IHC results, its representative images displayed a gradually increase in the number of RBP4-positive cells from C, O, P to OP with significant differences (Figure 6F). Apparently, an obese state could disrupt and impair the homeostasis of anti- and pro-inflammatory substances in adipokines in the periodontium, and triggered the inflammatory cascade in obesity-combined periodontitis.

### 2.6. Correlations of RBP4/Asprosin with Obesity and Periodontitis

After we confirmed the normality by checking all variables using the Shapiro–Wilk normality test, we then assessed the correlations of RBP4/asprosin mRNA expression in the gingiva with other clinical and biomarker variables of obesity and periodontitis using Spearman correlation (Table 1). Both RBP4 and asprosin were positively correlated (*p* < 0.0001) with the resorption level in CEJ-ABC, the gingiva mRNA expression of pro-inflammatory variables (IL-1β and TNF-α) and the osteoclastogenesis mediator RANKL with a strong correlation (*r* > 0.8). In addition, both RBP4 and asprosin were negatively correlated with the expression of the osteogenesis-related gene Col-1. Correlations between circulating indicator levels (asprosin, RBP4 and pro-inflammatory indictor IL-6) with RBP4 (or asprosin) mRNA expression in gingiva were each analyzed in order to ascertain the association between periodontally active sites and systematic expression. Interestingly, we found positive correlations between the circulating levels of the aforesaid cytokines (asprosin, RBP4 and IL-6) and gingiva RBP4 (or asprosin) mRNA expression, indicting a positive linear correlation in periodontal and systematic expression. Further, we identified a positive correlation between the RBP4 in the GCF level with RBP4 (or asprosin) mRNA expression (*r* > 0.8; *p* < 0.0001). We focused on the correlation analysis of RBP4 and asprosin with the clinical attachment level and the bone-remodeling related markers in detail (Figure 7). All these results infer a strong relationship between the expression of RBP4 and asprosin, and an increased level in these two adipokines may have an effect on or be accompanied by a higher degree of alveolar resorption, a decreased degree of osteogenesis and a higher level of pro-inflammatory variables.

## 3. Discussion

Periodontitis, a common disorder globally, is caused by periodontopathogens associated with multiple factors and could ultimately result in the irreversible destruction of the periodontium tissue [23]. The etiology of periodontitis is multi-factorial, and the subgingival dental biofilm during periodontitis could elicit a periodontal inflammatory and relevant immune response in a host. Obesity, which is regarded as a state of low-grade systemic inflammation [12], is also a risk factor for cardiovascular diseases, metabolic syndrome, diabetes, hypertension, and cancer (i.e., hepatocellular carcinoma and gallbladder) [24,25,26]. Obesity and related diseases could increase the risks of periodontal disease together with compromised periodontal healing, thus creating a complex network of associations. Among these, adipokines are speculated to be the pathomechanistic junctions between obesity and periodontitis [27]. Therefore, we demonstrated the patterns of adipokines involved in the pathogenesis of periodontitis and obesity.

Basically, consistent with the features of 2-week ligature-induced experimental periodontitis in male SD rats, our model was characterized by inflammation along with rapid alveolar bone resorption in the acute phase [28]. In addition, our ligature-induced model confirmed the distinguished bone resorption and the pro-inflammatory gene expression of IL-1β, TNF-α, CCL-2, Nos2, IL-17 and IL-6, which were highly enhanced during the progress of the early acute phase of new periodontitis [29]. With the appraisal of the successful establishment of an experimental obesity (significant differences in Lee’ s index as the gold standard and assisted by other biochemical variables) and periodontitis model, further analyses of adipokine pattern were more convincing for understanding the pivotal role of adipokines in co-morbidity.

Research has revealed that the highly increased expression level of Runx2 in the gingival tissue of patients with periodontitis may be associated with the pathogenesis of periodontitis [30,31]. It might be propitious to infer that during the dynamic process in bone remodeling, Runx2 was downregulated in the periodontitis for not propelling osteoblast differentiation. As the degree of periodontitis inflammation was cascaded by the obese status, the expression level of Runx2 in our study experienced a compensatory elevation for confronting or overcoming the negative effect of massive bone resorption and tissue reconstruction. And during periodontitis, the divergent expression profile driven by the transcriptional regulator Runx2 could be detected in macrophages, indicating the process of osteoblast differentiation led by Runx2 by directly regulating the Wnt, RANKL, FGF, hedgehog and Pthlh signaling molecules; these and other transcription factors under physiology condition were altered [31,32]. All these significant alternations in the expression levels of the key genes in our results implied that the continuous balance of bone resorption, mediated by osteoclasts, and bone formation, mediated by osteoblasts, in its entirety had been broken by the periodontitis and notably the dual periodontitis and obese status. Thus, the acute phase of periodontitis, which was characterized by inflammation and progressive bone resorption, was noted.

Firstly, our results of the adipokine pattern proved that leptin, visfatin, resistin, adiponectin, asprosin and RBP4 were also expressed at the periodontium, but not adipose tissue alone. Secondly, the gingiva mRNA expression of leptin, nampt/visfatin and resistin showed the maximum upregulation in the dual obese and periodontitis status, followed by in each single status, as compared with the control. Leptin is widely known for its function of inhibiting appetite, increasing energy consumption, and regulating the metabolism of adipose and bone tissue [33]. In addition, leptin is also expressed in osteoblasts and could work as a pro-inflammatory cytokine, participating in the host immune response [34]. The upregulated leptin in periodontitis with obesity might aggravate the progression of periodontitis and obesity via skewing proinflammatory M1 macrophage and stimulating proinflammatory cytokine expression in local periodontal ligament cells [35]. Visfatin could stimulate the synthesis of systematic inflammatory mediators and proteases in different types of cells while inhibiting the apoptosis of inflammatory cells. And it has been suggested as a pro-inflammatory biomarker and immunomodulator for periodontitis [36,37]. The expression of visfatin was proved to increase in gingival tissues of aggressive and periodontitis in human patients, as consistent with our results of the rat model [38]. Resistin, like the aforementioned two adipokines, could be released by adipocytes as well as inflammatory cells such as monocytes and macrophages [39]. The increasing level of resistin in obese patients [40] and rats with periodontitis might be due to its ability to recruit TNF-α and IL-6 [41]. As an antagonistic hormone of resistin, adiponectin is the only hormone with anti-inflammatory functions, accelerating the oxidation of fatty acids and so on. The organism’s self-protection mechanism towards the periodontium via promoting IL-10 and inhibiting the expression of ROS and other pro-inflammatory markers mediated by adiponectin [42] was suppressed in periodontitis rats with or without obesity in our study.

Previous studies investigating the relationship between the local and systematic expression of RBP4, asprosin and periodontitis, especially associated with the obese status, was rather equivocal, and this became the problem we focused on addressing. The periodontal mRNA and protein expression level of RBP4 in gingiva and GCF remained unchanged between the obese and control group. However, increased periodontal mRNA and protein expression levels of RBP4 in rats with periodontitis and the higher increased level in obese rats with periodontitis were found. As we know, RBP4 is a predictor of endothelial dysfunction, atherosclerosis and other cardiovascular diseases and is a member of the lipocalin family, which is bound to vitamin A and transthyretin while being secreted into the circulation. RBP4 is also greatly associated with inflammation and oxidative stress [43], and the circulating levels of RBP4 were highest in obese rats with periodontitis in our study. These findings about the GCF/serum secretion level of RBP4 were also consistent with the results in humans [10,44]. The underlying mechanisms might be that the occurrence of obesity and periodontitis would break the dynamic equilibrium between antioxidant defenses and ROS, which led to oxidative stress and the inflammatory response in the oral environment [45].

Considering another novel adipokine first described in 2016, asprosin is noted for inducing the synthesis of hepatic glucose, promoting islet β-cell inflammation, and inhibiting the secretion of insulin while raising glucose, and is regarded to be closely linked to obesity [9,46]. Our results demonstrated that asprosin was involved in the pathophysiological processes of periodontitis and obesity, such as the elevated local and systematic levels of asprosin accompanying the processes of periodontitis and obesity. Interestingly, the decreased circulating level of asprosin in obese rats of this study was inconsistent with those of previous studies [21,22] in obese adults and mice; however, it was consistent with some other studies in obese children [47,48]. This might be ascribed to the “honeymoon obese phase” [49] and the obviation of diabetes mellitus in our model. As a glucogenic and orexigenic hormone, asprosin might experience a compensatory reduction in circulating levels in order to maintain the homeostasis of glucose; this was also confirmed by the low HFD intake in the later period. However, this compensatory effect was offset by the comorbidity of periodontitis and obesity, and the circulating level was elevated thereof. Asprosin and obesity are deeply intertwined. HFD could stimulate the mRNA expression of asprosin in 3T3-L1 adipocytes and asprosin is able to induce the inflammatory response accompanied by the increased local and circulating level of inflammatory hallmarks, such as IL-6, TNF-a and so on [50]. As the expression level (both mRNA and protein) of asprosin was detected in periodontium, an especially higher level was found in rats with periodontitis and the highest level was found in rats with periodontitis and obesity, this may suggest asprosin also induces the inflammatory development of periodontal disease and worsen the comorbid condition [6]. Also, a positive correlation between the gingival mRNA expression of RBP4/asprosin and CEJ-ABC was observed, which indicates that higher gingiva RBP4/asprosin expression levels might point to an increased probability of periodontitis in both lean and obese populations. These findings imply that RBP4 and asprosin could be treated as novel biomarkers in the risk and healing assessment of complex periodontitis with obesity [10].

Above all, the underlying pathway in the pathogenesis of periodontitis and obesity might be ascribed to the disruption of or the imbalance in the expression of inflammatory mediators and ultimately contribute to the breakdown of periodontal supporting tissue. In the above, the initial increased pro-inflammatory adipokines and the decreased anti-inflammatory adipokines are greatly associated with oxidative stress both in local and systematic levels. And then a cascade of reactions take place in the inflammatory process of the next stage, in which the production of ROS and the adhesion of leukocyte–endothelial cells could be induced by Th17/IL-17 and macrophages via the secretion of TNF-α, IL-6 and so on. The upregulated expression of hallmarks for inflammation, and the corresponding degradation in connective tissue and the generation of osteoclasts in periodontitis are activated and aggravated in obesity.

## 4. Materials and Methods

### 4.1. Animals and Experimental Design

All animal operation procedures were approved by the Xi’an Jiaotong University Ethical Committee on the Use of Care of Animals in respect to Animal Experimentation (XJTUAE2023-1285). SPF Laboratory male SD rats, with an initial body weight of 150–200 g, were obtained from the Medical Experimental Animal Center of Xi’an Jiaotong University. These animals were maintained under a 22–25 °C controlled temperature with a 12 h light/dark cycle, and received a standard laboratory diet (CHOW-diet, the calorie content of which was 13.5% fat, 58% carbohydrates, and 28.5% protein; 5001, LabDiet, Texas, USA) and water ad libitum for 1 week to adapt to the environment. Then, the littermate SD rats with a similar age and weight were divided into four groups (n = 8) by random allocation (random number method): (1) no treatment as the control group, receiving a CHOW-diet (C); (2) an obesity group receiving a high-fat diet (HFD) (O); (3) a ligature-induced periodontitis group receiving a CHOW-diet (P); (4) and a ligature-induced periodontitis-associated-with-obesity group receiving a HFD (OP). From week 2 to week 12, two groups of SD rats received a CHOW-diet or high-fat diet (HFD, the calorie content of which was 60% fat, 20% carbohydrates and 20% protein; Beijing Keao, Research Diets, Beijing, China). Body weight was recorded once every three days. And at the end of week 12, the difference in body mass between groups with or without the induction of obesity needed to be at least 15% [51]; otherwise, the induction time was prolonged.

An experimental periodontitis model was built through ligaturing the maxillary second molar, and the silk ligatures were pre-soaked in lipopolysaccharides for 24 h (LPS-PG, Lipopolysaccharide from *Porphyromonas gingivalis*, cat. code: tlrl-pglps, 1 mg/mL, InvivoGen, San Diego, CA, USA) and checked for retention every two days. The whole ligature-induced periodontitis period lasted for 2 weeks before each group of animals was sacrificed at the same timepoint.

### 4.2. Metabolic Studies

All animals were weighed every two days. The food intake per day and Lee’s index in the different groups were measured. At the end of 12-week, rats were fasted for 8 h and then subjected to decapitation treatment after general anesthesia, and FBGL were measured using a handheld blood glucose meter. Blood was collected with heparin sodium and centrifugated (2700× *g*, 15 min) at 4 °C to obtain plasma. Then, the plasma was analyzed for triglyceride, total cholesterol, LDL-C and HDL-C using the matched assay kits (A110-1-1, A111-1-1, A113-1-1 and A112-1-1; Nanjing Jiangcheng Bioengineering Institute, Nanjing, China). Furthermore, the subcutaneous and retroperitoneal peripheral adipose tissue, epididymal visceral adipose tissue and BAT of rats were dissected bilaterally and weighed for body composition analysis.

### 4.3. Micro-Computed Tomography

The hemi-maxillae were dissected and fixed in 4% paraformaldehyde for 48 h. Then, the samples were stored in 70% ethanol before being scanned by a high-resolution small animal vivo micro-CT system (Quantum GX; Bruker SkyScan micro-CT, PerkinElmer, MA, USA). Further, raw images of the teeth were reconstructed and captured using standard SkyScan reconstruction software (NRecon, version 1.7.1.0; Bruker) in three simultaneous dimensions, with a special focus on the buccolingual and mesiodistal apical–coronal directions of the maxillary second molar. Moreover, 3D-conebeam reconstruction algorithms were utilized to reconstruct and align the 3D images of each whole sample. All images were analyzed using the CT-Analyzer software (SkyScan; Bruker micro-CT, version: Analyze12.0). Linear bone loss (LBL) was measured via calculating the distance from the CEJ to the alveolar bone crest (ABC) in the buccolingual and mesiodistal apical–coronal directions of the maxillary second molar, and all 3D images were captured with the same angle for intuitive comparison.

### 4.4. Histological Evaluation

Other lateral hemi-maxillae samples of rats in four groups were fixed in 4% paraformaldehyde for sufficient decalcification and dehydration. Fixed samples were embedded in paraffin wax, and sliced into a series of 4 µm thick mesiodistal sections of the second molar. H&E staining and TRAP staining were then performed. Microphotographs of the mesial, distal, and furcation areas focused on the second molar were taken at 200× magnification by a digital camera (DP80, Olympus, Tokyo, Japan) and were latterly measured and analyzed using the Image J software (version 1.8.0). The distribution of inflammatory cells was marked in images of H&E staining by the experienced examiner blinded to the experimental groups. Osteoclasts, also seen as TRAP-positive cells, were marked and quantified by the same examiner.

### 4.5. Immunohistochemistry

Immunohistochemistry was performed on the same serial 4 µm thick sections, which were incubated with each primary antibody anti-RBP4 (11774-1-AP, Proteintech, Wuhan, China) and asprosin (FNab09797, FineTest, Wuhan, China). Stainings were visualized by an ABC Kit (KIT-9730, Maixin, Shenzhen, China) and DAB substrate (DAB-1031, Maixin, Shenzhen, China). Sections were also counter-stained with hematoxylin for nuclei visualization. The eventual microphotographs of the mesial, distal, and furcation areas focused on the second molar were observed and captured at 200× magnification under a light microscope (BX-51, Olympus, Tokyo, Japan), and were latterly measured and analyzed by the Image J software (version 1.8.0) by counting the positive cells by the same researcher blinded to the group distribution.

### 4.6. Quantitative Real-Time PCR Analysis

The gingival tissue of the ligated and non-ligated maxillary second molar on the counterparts were dissected and frozen in liquid nitrogen, then stored at −80 °C. Total RNA was then isolated and quantified from the gingival tissue via the TRIzol RNA isolation reagents (Thermo Scientific, Waltham, MA, USA). cDNA was reversed from the quantified RNA utilizing a reverse transcription kit (Thermo Scientific, Waltham, MA, USA). Quantitative real-time PCR analysis was proceeded by a Bio-Rad 9060 thermocycler with 2X SYBR Green Pro Taq HS Premix qPCR kit (Accurate Biology, Shenzhen, China). Eventually, relative mRNA expressions of the interested genes were determined using β-actin as the housekeeping gene for normalization by using the 2 ^−ΔΔCt^ method. Focused genes: IL-1β; IL-6; CCL-2; TNF-α; Nos2; IL-17A; RBP4; asprosin; leptin; Nampt (visfatin); Resistin (Retn); adiponectin (Adipoq); RANKL; ALP; Runx2; Pthlh; Ctsk; and Col-1. The related primers were synthesized using AUGCT and are presented in the Appendix A, Beijing, China).

### 4.7. Enzyme-Linked Immunosorbent Assay

GCF was collected before sacrificing using filter paper strips (Whatman1#, OCOME, Hangzhou, China) gently placed into the bottom of the gingival pocket at the proximal and distal sites for 30 s. And the strip tops were then cut and placed in PBS solution of 150 µL for dissolution for about 1 h. After centrifugation (11,500× *g* at 4 °C for 10 min), the supernatant was extracted. RBP4/IL-6 levels in the GCF and plasma were determined using rat ELISA kits (ER0395 and ER0042, FineTest, Wuhan, China). Meanwhile, plasma asprosin levels were measured using a rat asprosin ELISA kit (ER1944, FineTest, Wuhan, China). Absorbance was detected using an Epoch microplate reader (BioTek, Winooski, VT, USA) at 450 nm according to the manufacturers’ instructions.

### 4.8. Western Blot Analysis

For extracting total proteins from the gingival tissue, a homogenizer with a radioimmunoprecipitation assay (RIPA) lysate (Beyotime, Nantong, China) was utilized. The concentration of protein samples was determined using the bicinchoninic acid (BCA) protein assay kit (Thermo Scientific, Waltham, MA, USA). Protein samples were therefore separated in SDS-PAGE gels (BioRad, Hercules, CA, USA), then transferred to Polyvinylidene difluoride (PVDF) membranes (Millipore, Bedford, MA, USA). Next, the membranes were blocked with NcmBlot speedy blocking buffer for 10 min at room temperature. Later, the membranes or strips were incubated with each primary antibody: IL-1β (YT5201, Immunoway, Plano, TX, USA), RBP4 (11774-1-AP, Proteintech, Wuhan, China), asprosin (FNab09797, FineTest, Wuhan, China), fibrillin 1 (YT1684, FBN1, Immunoway, Plano, TX, USA) and β-actin (P30002F, Abmart, Shanghai, China) (all diluted 1:2000), overnight at 4 °C, and the second antibody (RS002, Immunoway, Plano, TX, USA, anti-rabbit, 1:10,000) in the ambient for 1 h after being washed with Tris-buffered saline and Tween (TBS-T). Protein bands were analyzed using a transilluminator and imaging analysis system (Bio-Rad, Hercules, CA, USA) with NcmECL imaging while the densities of which were calculated using Quantity One software version 4.3.0 (BioRad, ChemiDocTM Touch Imaging System, Hercules, CA, USA).

### 4.9. Statistical Analysis

For all the evaluated parameters involved in the experiments of this study, analyses were implemented at least twice with 3–8 replicates, and the data are presented as the mean ± standard error of the mean (SEM). The values of *p* < 0.05 represented differences between groups were statistically significant.

Normality was confirmed after checking all variables through the Shapiro–Wilk normality test. Statistical significances among multiple groups were evaluated by one-way ANOVA. Differences among groups were compared with the Fisher post hoc least significant difference test, and Spearman’s correlation coefficient was utilized for data correlation analysis. GraphPad Prism Software version 9.4.0 (GraphPad Software, La Jolla, CA, USA) and SPSS version 22 (SPSS Inc., Chicago, IL, USA) were used for the statistical analyses in this study.

## 5. Conclusions

To sum up, as far as we are concerned, this was the first study to investigate the role of RBP4 and asprosin in periodontitis associated with obesity. The present study revealed a divergent adipokine expression profile, and notably a strong relationship between the expression of RBP4 and asprosin and the increased periodontal and circulating levels of RBP4 and asprosin in experimental periodontitis and diet-induced obese rats. The increased level of RBP4/asprosin may have an effect on the more serious alveolar bone resorption, the less active osteogenesis and the higher level of pro-inflammatory variables. The potential pathway in the pathogenesis of periodontitis and obesity might be ascribed to the disruption of or the imbalance in the expression of inflammatory mediators, and ultimately contribute to the breakdown of periodontal supporting tissue. In the above, the upregulated expression of hallmarks for inflammation in periodontitis were activated and exacerbated in obesity by adipokines, especially Rbp4 and asprosin. The aforementioned findings provide perceptions of the underlying mechanisms engaged in periodontitis in obesity and expound these novel biomarkers, respectively, as potential candidates for pragmatic clinical therapies.

## Figures and Tables

**Figure 1 ijms-24-16739-f001:**
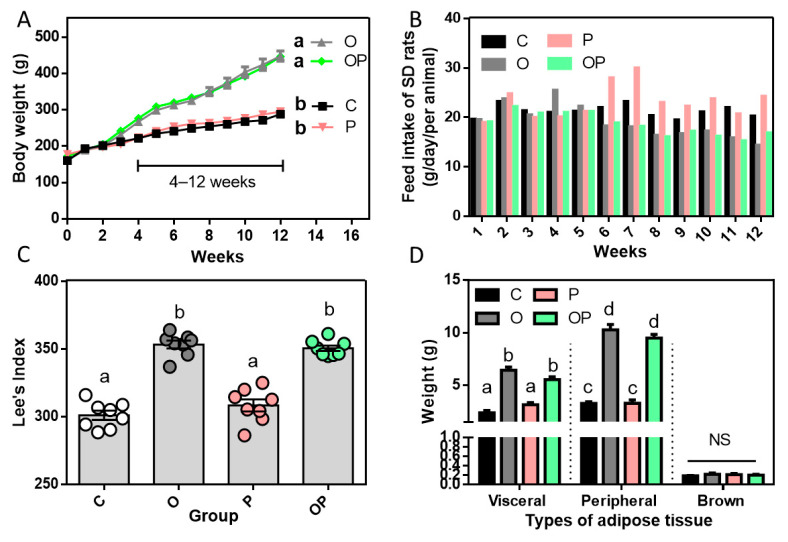
Analyses of obese SD rat model induced by HFD. (**A**) Changes in body weight during the 12 weeks; significant difference was detected over the period from week 4 to 12 between obesity group receiving HFD and those receiving CHOW diet. (**B**) Feed consumption of one rat per day during the 12 weeks. (**C**) Lee’s index of each group. (**D**) Weight of visceral and peripheral adipose tissue in WAT along with BAT. Data are presented as mean ± SEM, n = 8. Different letters indicate statistical differences among groups.

**Figure 2 ijms-24-16739-f002:**
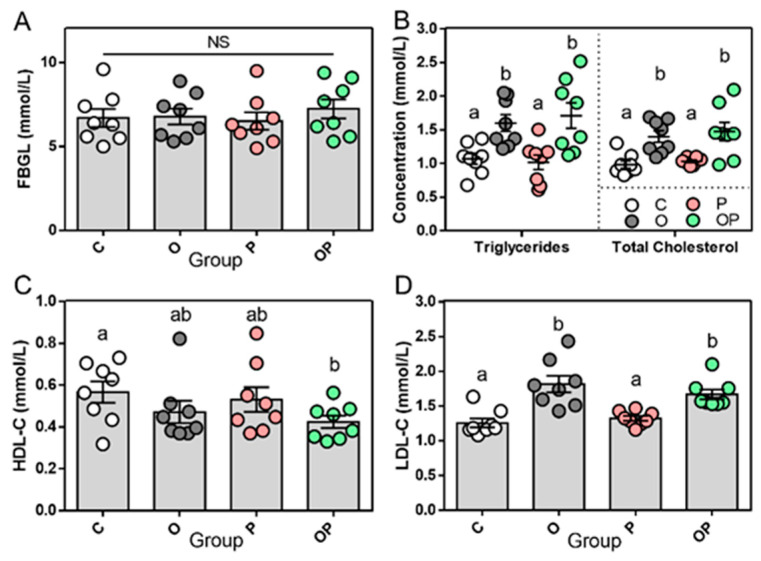
Biochemical analyses of established obese rat model. (**A**) Levels of fasting blood glucose. The concentrations of triglycerides, total cholesterol (**B**), HDL-C (**C**) and LDL-C (**D**) in plasma. Data are presented as mean ± SEM, n = 8. Different letters indicate statistical differences among groups.

**Figure 3 ijms-24-16739-f003:**
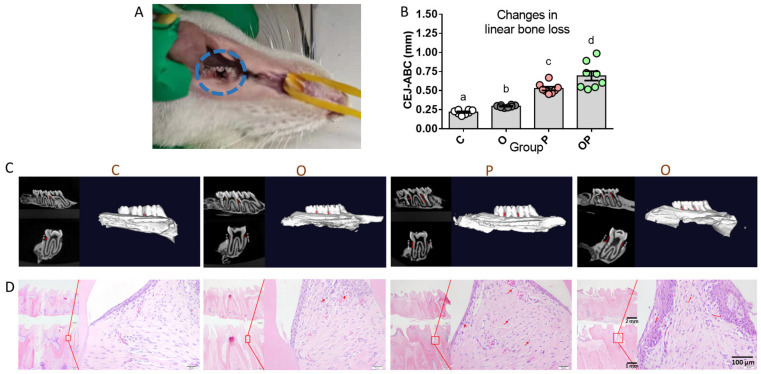
Analyses of 2-week ligature-induced periodontitis rat model. (**A**) Image of ligature-induced periodontitis model. (**B**) Quantitative study of linear resorption distances from CEJ to ABC. (**C**) Representative images of specimens demonstrating bone resorption in four groups of C, O, P and OP. (**D**) H&E-stained sections from four groups with the marked infiltration of inflammatory cells (red arrowheads). Data are presented as means ± SEM, n = 8. Different letters indicate statistical differences among groups. H&E, hematoxylin and eosin; CEJ, cementoenamel junction; ABC, alveolar bone crest.

**Figure 4 ijms-24-16739-f004:**
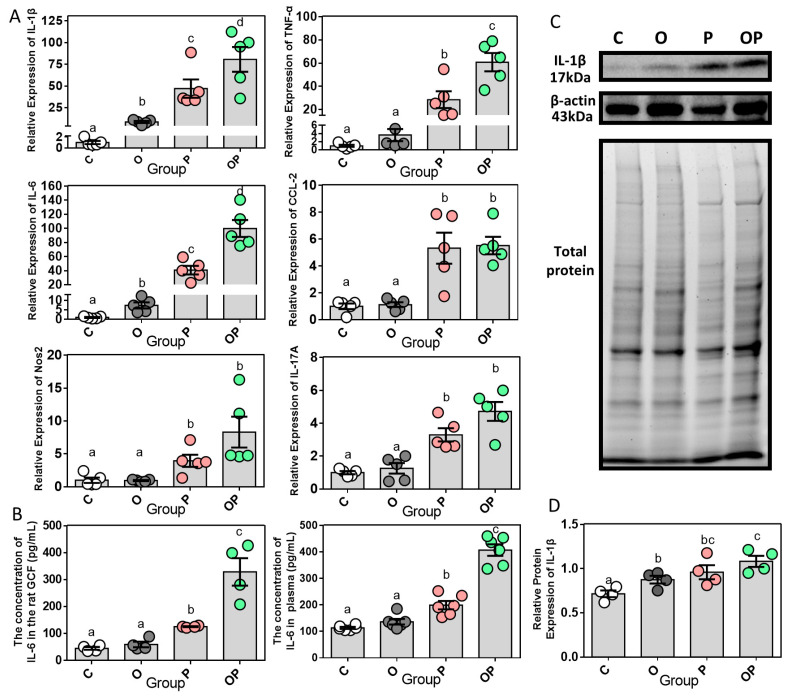
Expression of periodontal and systemic inflammatory cytokines. (**A**) Relative inflammatory gene expression in gingiva, n = 5. (**B**) ELISA analysis of IL-6 in GCF (n = 4) and plasma (n = 6). (**C**) WB analysis of IL-6 protein expression in gingiva, n = 4. (**D**) Quantification of Western blot analysis. Protein content is expressed relative to the control and represents three independent experiments, with triplicate observations in each experiment. The volume is the sum of all pixel intensities within a band. All data are normalized to β-actin and are expressed as mean ± SEM. Different letters indicate statistical differences between groups.

**Figure 5 ijms-24-16739-f005:**
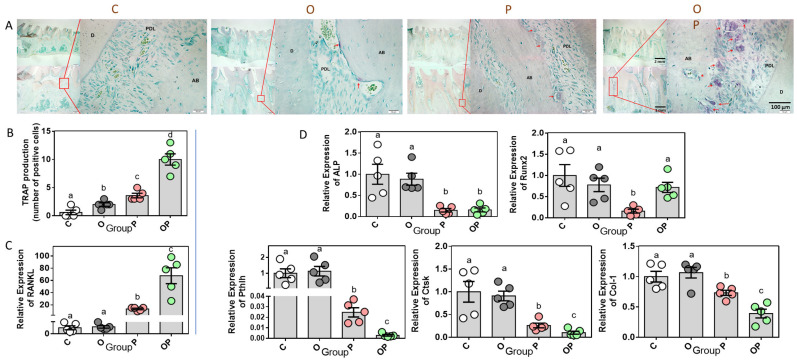
Expression of bone remodeling biomarkers in periodontium. Representative images (**A**) and quantitative analysis (**B**) of periodontium tissue sections stained for TRAP from the four groups. TRAP-positive cells/osteoclasts are indicated by red arrowheads. Relative bone remodeling-related gene expression in gingival tissue, with osteoclastic-related (**C**) and bone formation-related (**D**) gene expression. Data are presented as means ± SEM, n = 5. Different letters indicate statistical differences among groups. D, dentin; AB, alveolar bone; PDL, periodontal ligament.

**Figure 6 ijms-24-16739-f006:**
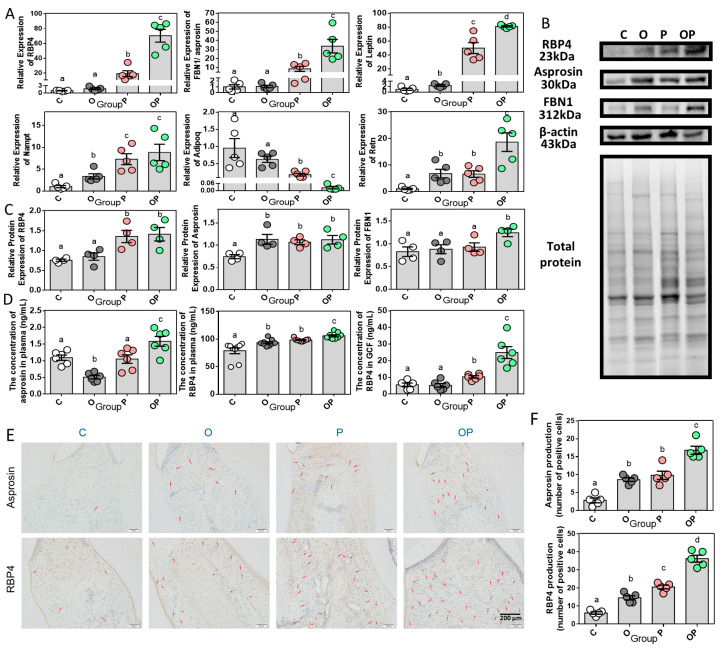
Periodontium and systematic expression of adipokines in obesity and periodontitis. (**A**) Relative adipokine gene expression in gingiva, n = 5. (**B**) WB analysis (n = 4) of representative RBP4, asprosin and FBN1 protein expression in gingiva. (**C**) Quantification of Western blot analysis. Protein content is expressed relative to the control and represents three independent experiments with triplicate observations in each experiment. Volume is the sum of all pixel intensities within a band. All data are normalized to β-actin and are expressed as mean ± SEM. (**D**) ELISA analysis of asprosin in plasma and RBP4 in GCF and plasma, n = 6. (**E**) Representative immunohistochemistry images of periodontium tissue sections for asprosin and RBP4 from four groups. Positive cells are indicated by red arrowheads. (**F**) Quantification of asprosin and RBP4-positive cell of the four groups (data were collected from immunohistochemistry images of five individual rats in each group). Data are presented as mean ± SEM. Different letters indicate statistically significant differences among groups. Adipoq, adiponectin; Retn, resistin.

**Figure 7 ijms-24-16739-f007:**
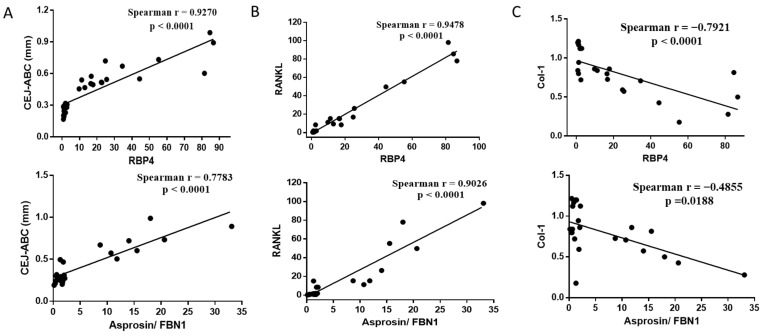
Correlation analysis of RBP4 and asprosin with clinical attachment level and the bone-remodeling-related markers of the study groups. Partial regression plots of gingiva RBP4/asprosin mRNA expressions with the relative levels of clinical attachment level (**A**), RANKL (**B**) and Col-1 (**C**) mRNA expressions, respectively. All participants from the four groups (C, O, P and OP) were analyzed for correlation. (n = 20); the Pearson correlation analysis was used as the statistical test.

**Table 1 ijms-24-16739-t001:** Correlation analysis between RBP4/asprosin and clinical and biomarker data in all participants.

	RBP4 ^a^		Asprosin ^a^	
	*r*	*p*	*r*	*p*
CEJ-ABC	0.9270	<0.0001	0.7783	<0.0001
IL-1β ^a^	0.9610	<0.0001	0.8322	<0.0001
TNF-α ^a^	0.9278	<0.0001	0.9455	<0.0001
RANKL ^a^	0.9478	<0.0001	0.9026	<0.0001
Col-1 ^a^	−0.7921	<0.0001	−0.4855	0.0188
Asprosin ^b^	0.4555	0.0289	0.7144	0.0001
RBP4 ^b^	0.8783	<0.0001	0.8417	<0.0001
IL-6 ^b^	0.9206	<0.0001	0.7775	<0.0001
RBP4 ^c^	0.8550	<0.0001	0.8052	<0.0001

Spearman’s correlation was used. ^a^ mRNA expression in gingiva; ^b^ Concentration in plasma; ^c^ Concentration in GCF.

## Data Availability

The datasets used and/or analyzed during the study are available from the corresponding author upon reasonable request.

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
