# Peer review of "Increased RBP4 and Asprosin Are Novel Contributors in Inflammation Process of Periodontitis in Obese Rats"

_ijms, 2023, doi:10.3390/ijms242316739_

Round 1

Reviewer 1 Report

Comments and Suggestions for Authors

Zang et al tested in vivo the innovative hypothesis that few adipokines enhanced inflammation in obese rats in the experimental model of periodontitis.

From what I understood, the major aim was to test expression levels of cytokines and adipokines as well as analyze involved intracellular signaling pathways.

Moreover, the authors describe aprosin and RBP4 as novel adipokines manipulating inflammatory responses in obese, periodontal rats, which might serve as a therapeutic target for periodontitis patients in the future.

. I believe the science is well conducted with all appropriate controls. I have only minor issues with agreeing with the authors on their interpretations.

Issues:

1.         Regarding the Methodology (Lane 87), please provide detailed information on LPS type, from which organism it was isolated, catalog number, and concentration.

2.         Figure 3C, 4A, 5B,C,D; 6A,B,C, - could you magnify the graphs? It isn't easy to read the X and Y axis markings.

3.         Have the authors considered the analysis of IL-17 expression? It has been recently reported to be a pro-inflammatory interleukin involved in periodontitis.

4.         Regarding Figure 4, protein loading (Actin)on Western Blot seems uneven. Do the authors have another experiment to show?

Author Response

  1. Regarding the Methodology (Lane 87), please provide detailed information on LPS type, from which organism it was isolated, catalog number, and concentration.

Response: Thank you so much for the kind encouragement and valuable comment, which really mean a lot to us. We have added the information in the manuscript as “Experimental periodontitis model was built through ligaturing the maxillary second molar, and the silk ligatures were pre-soaked in lipopolysaccharides for 24 hours (LPS-PG, Lipopolysaccharide from Porphyromonas gingivalis, cat. code: tlrl-pglps, 1mg/mL, InvivoGen, San Diego, CA, USA) and checked for retention every two days.”. Please read the details in the revised version (page 2, line 95-97).

  1. Figure 3C, 4A, 5B, C, D; 6A, B, C, - could you magnify the graphs? It isn't easy to read the X and Y axis markings.

Response: Thank you very much for your constructive reminding. We have magnified the graphs and the indistinguishable axis markings for better readability according to your advice. Please read the details in the revised version.

  1. Have the authors considered the analysis of IL-17 expression? It has been recently reported to be a pro-inflammatory interleukin involved in periodontitis.

Response: Thank you for the supplied information that keeps pace with the latest trend of research. Currently, our research on the inflammatory factors is mainly aimed at verifying the establishment of periodontitis models, so we have selected some relatively classic inflammatory factors. However, your suggestion is actually a good reminder for us, so we have detected the gene expression of IL-17 in gingiva of the four groups according to the reviewer’s advice. And we found it was highly enhanced during the progress of early acute phase of periodontitis in NP and OP groups. We have shown this data in our revised Figure 4 and modified the result part based on this new observation and reviewer’s suggestions. Please find the details in the revised version (page 7 line 277).

  1. Regarding Figure 4, protein loading (Actin)on Western Blot seems uneven. Do the authors have another experiment to show?

Response: Many thanks for your kind suggestions. In fact, as shown in Figure 4, when we quantified the expression level of the proteins, we not only referred to the internal reference, but also homogenized the total protein quantification of all swim lanes. Therefore, we think that even if the internal reference expression levels are not completely consistent, we can still obtain relatively objective quantitative results of protein expression levels. However, we completely agree with you that it will be better to show another result of the experiment here. We have changed another group of experimental gels with relatively even protein loading (Actin) in the revised Figure 4. Please read the details in the revised version.

In the end, we would like to express our special thanks to you for all your comments and suggestions, which are of great help to improve the quality of our manuscript!

Reviewer 2 Report

Comments and Suggestions for Authors

Congratulations, this document demonstrates a huge amount of work and is very well structured. Nevertheless, I think that there is too much information for a single paper. I recommend dividing this paper in two, in order to facilitate the understanding and interest of the reader. In that way, the discussion can be improved by dealing with more specific issues. 

The actual title could be maintained for one of the articles.

Animal and experimental design should be reviewed as it is not clear the way of randomization and how the animals were distributed by the 4 groups.

Author Response

Responses to the reviewers’ comments:

  1. Congratulations, this document demonstrates a huge amount of work and is very well structured. Nevertheless, I think that there is too much information for a single paper. I recommend dividing this paper in two, in order to facilitate the understanding and interest of the reader. In that way, the discussion can be improved by dealing with more specific issues. The actual title could be maintained for one of the articles.

Response: Thank you so much for your sincere encouragement and constructive comment, which really mean a lot to us. We think that such structure and arrangement is conducive to presenting the entire research ideas and content to readers, thereby facilitating the derivation of conclusions. We understand that due to the much information of the content, the Discussion section may be somewhat "chaotic" and not compact enough. We have rechecked and made relevant adjustments accordingly to improve our manuscript. Please refer to our revised version for details (page 12-16).

  1. Animal and experimental design should be reviewed as it is not clear the way of randomization and how the animals were distributed by the 4 groups.

Response: We deeply appreciate your insightful and professional comments. we have modified the manuscript as follows. “All animal operation procedures were approved by the Xi'an Jiaotong University Ethical Committee on Use of Care of Animals in respect to Animal Experimentation (XJTUAE2023-1285). SPF Laboratory male Sprague Dawley rats (SD rats), with the initial body weight of 150-200 g, were obtained from Medical Experimental Animal Center of Xi'an Jiaotong University. These animals were maintained under 22–25 °C controlled temperature with a 12-hour light/dark cycle, and received a standard laboratory diet (CHOW-diet, the calorie content of which is 13.5% fat, 58% carbohydrates, and 28.5% protein, Texas, LabDiet 5001) and water ad libitum for 1 week to adapt to the environment. Then, the littermates SD rats with similar age and weight were divided into four groups (n = 8) by random allocation (random number method), as: (1) no treatment as control group receiving a CHOW-diet (C); (2) obesity group receiving a high-fat diet (HFD) (O); (3) ligature-induced periodontitis group receiving a CHOW-diet (P); (4) and ligature-induced periodontitis associated with obesity group receiving a HFD (OP). From week 2 to week 12, two groups of SD rats were received a CHOW-diet or high-fat diet (HFD, the calorie content of which is 60% fat, 20% carbohy-drates and 20% protein, Beijing Keao, Research Diets). Body weight was recorded once every three days. And at the end of week 12, the difference in body mass between groups with or without the induction of obesity need to be at least 15% [11], otherwise the induction time should be prolonged.”. We hope this will show the animal and experimental design more clearly. Please read the details in the revised version (page 2, line 71-88).

In the end, we would like to express our special thanks to you for all your comments and suggestions, which are of great help to improve the quality of our manuscript!

Reviewer 3 Report

Comments and Suggestions for Authors

Dear Authors,

The article entitled "Increased RBP4 and asprosin are novel contributors in inflammation process of periodontitis in obese rats" investigated the expression profile and possible role of RBP4, asprosin and other adipokines in experimental periodontitis and diet-induced obese rats.

The introduction chapter is well-written and emphasizes the problem.

The Materials and Methods chapter describes the experiment design. 

The results are well-exposed and original. I recommend the enlargement of the images included in Figures 3 and 5 only to increase the visibility of the differences.

The Discussion chapter is well-presented. You can add the idea that obesity is also a risk factor for cardiovascular diseases, diabetes, hypertension, and cancer. These also contribute to periodontal disease.

Here are 2 recommendations to improve the manuscript.:

Line 427 Replace "except for" with "but not".

Line 507 in the periodontitis under obesity Replace "under" with "associated with"

I recommend a minor revision of the manuscript.

Author Response

Responses to the reviewers’ comments:

The article entitled "Increased RBP4 and asprosin are novel contributors in inflammation process of periodontitis in obese rats" investigated the expression profile and possible role of RBP4, asprosin and other adipokines in experimental periodontitis and diet-induced obese rats. The introduction chapter is well-written and emphasizes the problem. The Materials and Methods chapter describes the experiment design.

  1. The results are well-exposed and original. I recommend the enlargement of the images included in Figures 3 and 5 only to increase the visibility of the differences.

Response: Thank you very much for your constructive reminding. We have magnified the indistinguishable axis markings for better readability according to your advice. Please read the details in the revised Figure 3 and 5 in page 7 and 9.

  1. The Discussion chapter is well-presented. You can add the idea that obesity is also a risk factor for cardiovascular diseases, diabetes, hypertension, and cancer. These also contribute to periodontal disease.

Response: Thank you so much for your sincere suggestion, it is so helpful and we have added the corresponding views in the updated version as “Periodontitis, as a common disorder globally, is caused by periodontopathogens associated with multi-factors and could ultimately result in the irreversible destruction of the periodontium tissue [24]. The aetiology of periodontitis is multi-factorial, the subgingival dental biofilm during periodontitis could elicit a host periodontal inflammatory and relevant immune response. Obesity, which regarded as a state of low-grade systemic inflammation [13], is also a risk factor for cardiovascular diseases, metabolic syndrome, diabetes, hypertension, and cancer (i.e., hepatocellular carcinoma and gallbladder) [25-27]. Obesity and the related diseases could increase the risks of periodontal disease together with the compromised periodontal healing, thus creating a complex network of associations. Among in, adipokines are speculated to be the pathomechanistic junctions between obesity and periodontitis [28]. Therefore, we demonstrated the pattern of adipokine involved in the pathogenesis of periodontitis and obesity.”. Thank you for bringing this oversight to our attention. Please read the details in the revised version (page 13, line 413-423).

Here are 2 recommendations to improve the manuscript.:

  1. Line 427 Replace "except for" with "but not".

Response: We appreciate the reviewer for the constructive suggestion. We have replaced " except for " with " but not " in this revision (page13, line455).

  1. Line 507 in the periodontitis under obesity Replace "under" with "associated with"

Response: Thank you so much for your valuable suggestion. We have replaced "under" with "associated with" in this revision (page15, line 536).

Thank you very much again for the constructive comments and suggestions. We have tried our best to make as many modifications to our manuscript as possible according to your suggestions. We believe these modifications have greatly improved the quality of our manuscript without significantly impacting the major content, framework and conclusions we made in this paper. We deeply appreciate the professional comments and hard work the editors and reviewers have made to our work. We hope that the modifications and improvements we have made will meet your standards of satisfaction.

Round 2

Reviewer 2 Report

Comments and Suggestions for Authors

No further comments